# Non-Obese MAFLD Is Associated with Colorectal Adenoma in Health Check Examinees: A Multicenter Retrospective Study

**DOI:** 10.3390/ijms22115462

**Published:** 2021-05-22

**Authors:** Shuhei Fukunaga, Dan Nakano, Takumi Kawaguchi, Mohammed Eslam, Akihiro Ouchi, Tsutomu Nagata, Hidefumi Kuroki, Hidemichi Kawata, Hirohiko Abe, Ryuichi Nouno, Koutaro Kawaguchi, Jacob George, Keiichi Mitsuyama, Takuji Torimura

**Affiliations:** 1Division of Gastroenterology, Department of Medicine, Kurume University School of Medicine, 67 Asahi-machi, Kurume 830-0011, Japan; nakano_dan@med.kurume-u.ac.jp (D.N.); takumi@med.kurume-u.ac.jp (T.K.); ohuchi_akihiro@med.kurume-u.ac.jp (A.O.); nagata_tsutomu@med.kurume-u.ac.jp (T.N.); ibd@med.kurume-u.ac.jp (K.M.); tori@med.kurume-u.ac.jp (T.T.); 2Storr Liver Centre, Westmead Institute for Medical Research, Westmead Hospital and University of Sydney, 176 Hawkesbury Rd, Westmead, NSW 2145, Australia; mohammed.eslam@sydney.edu.au (M.E.); jacob.george@sydney.edu.au (J.G.); 3Diagnostic Imaging Center, Kurume University Hospital, 67 Asahi-machi, Kurume 830-0011, Japan; kuroki_hidefumi@kurume-u.ac.jp (H.K.); kawata_hidemichi@kurume-u.ac.jp (H.K.); 4Department of Gastroenterology, Kumamoto Central Hospital, 2921 Haramizu, Kikuyo, Kikuchi 869-1102, Japan; aaa@kchosp.or.jp (H.A.); r-nono@kchosp.or.jp (R.N.); 5Kurate Hospital, 2425-9 Nakayama Kurate-machi, Kurate 807-1312, Japan; kotaro481108@yahoo.co.jp

**Keywords:** colorectal neoplasms, metabolic syndrome, non-alcoholic fatty liver disease, thinness, liver steatosis

## Abstract

Colorectal adenoma is linked to metabolic dysfunction. Metabolic dysfunction-associated fatty liver disease (MAFLD) has a precise definition and three subtypes, including non-obese MAFLD. We aimed to investigate the impact of MAFLD on the prevalence of colorectal adenoma by comparing it to non-alcoholic fatty liver disease (NAFLD) in health check-up examinees. This is a multicenter retrospective study. We enrolled 124 consecutive health check-up examinees who underwent colonoscopy. NAFLD and MAFLD were present in 58 and 63 examinees, respectively. Colorectal adenoma was diagnosed by biopsy. The impact of the MAFLD definition on the prevalence of colorectal adenoma was investigated by logistic regression, decision-tree, and random forest analyses. In logistic regression analysis, MAFLD was identified as the only independent factor associated with the presence of colorectal adenoma (OR 3.191; 95% CI 1.494–7.070; *p* = 0.003). MAFLD was also identified as the most important classifier for the presence of colorectal adenoma in decision-tree and random forest analyses (29 variable importance value). Among the three subtypes of MAFLD, non-obese MAFLD was the sole independent factor associated with the presence of colorectal adenoma (OR 3.351; 95% CI 1.589–7.262; *p* ≤ 0.001). Non-obese MAFLD was also the most important classifier for the presence of colorectal adenoma in decision-tree and random forest analyses (31 variable importance value). MAFLD, particularly non-obese MAFLD, is the most important factor associated with the presence of colorectal adenoma rather than NAFLD. Colonoscopy examination should be considered in patients with MAFLD, especially those who are non-obese.

## 1. Introduction

The prevalence of colorectal cancer is increasing, with over 1.8 million new colorectal cancer cases being reported each year [1]. Colorectal cancer ranks second in terms of mortality, and 881,000 deaths were estimated to have occurred in 2018 [1]. Most colorectal cancers develop through the adenoma-carcinoma sequence [2]. A multicenter post-polypectomy surveillance study showed that colonoscopic polypectomy significantly reduces the risk of death from colorectal cancer [3]. Thus, treatment of colorectal adenoma is likely to be an effective strategy to reduce the risk of death from this cancer. This decision can be informed by assessing the risk factors associated with colorectal adenoma.

Aging, gender, and family history increase the risk of colorectal cancer; however, these factors are unchangeable [4]. Many lifestyle-related factors have also been linked to colorectal cancer. These factors are modifiable, and meta-analyses demonstrate that high-risk factors for colorectal cancer are smoking, alcoholic intake, and metabolic dysfunction such as obesity, type 2 diabetes mellitus (T2DM), hypertension, and dyslipidemia [5,6]. NAFLD increases the risk not only for hepatocellular carcinoma but also of extrahepatic cancers including colorectal cancer [7]. Recent studies also showed that NAFLD is associated with an increased risk of colorectal adenoma and screening colonoscopy is recommended for patients with NAFLD [8,9]. However, NAFLD is a heterogeneous disease and can be diagnosed irrespective of the presence of metabolic dysfunction.

Recently, a panel of experts from 22 countries proposed a new definition for the diagnosis of metabolic dysfunction-associated fatty liver disease (MAFLD) [10]. The diagnosis of MAFLD is based on evidence of fatty liver in addition to one of (1) overweight/obesity, (2) the presence of T2DM, or (3) presence of metabolic dysregulation with at least two risk features including central obesity, pre-diabetes, hypertension, hyperlipidemia, and depressed high-density lipoprotein (HDL) cholesterol [10]. Accordingly, MAFLD, rather than NAFLD, is likely to be more associated with metabolic dysregulation-related events. In fact, MAFLD is reported to identify patients with significant hepatic fibrosis better than NAFLD [11,12]. However, it remains unclear whether MAFLD is superior to NAFLD as a factor associated with colorectal adenoma.

The aim of this study is to investigate the impact of MAFLD on colorectal adenoma by comparing it to NAFLD in health check-up examinees.

## 2. Results

### 2.1. Patients’ Characteristics

In all subjects, the median age was 59 years and the male ratio was 80.6% (Table 1). The median body mass index was 23.1 and the prevalence of fatty liver was 58.1% (72/124) of subjects. Severe fatty liver (fatty liver index >60) was seen in 19% and advanced hepatic fibrosis (NAFLD fibrosis score > 0.675, corresponding to F3-F4) was seen in 2% (Table 1). In all subjects, 37.9% were diagnosed with colorectal adenoma (Table 1).

NAFLD and MAFLD were present in 80.6% and 87.5% of subjects with fatty liver (*n* = 72), respectively. Patients overlapping NAFLD and MAFLD comprised 69.4% (50/72) of the subjects with fatty liver (Figure 1).

Patients’ characteristics are summarized in Table 1. There was no significant difference in age, sex, and body mass index (BMI) between the NAFLD and MAFLD groups. In the MAFLD group, the prevalence of alcohol drinkers (men ≥30 gms/day, women ≥20 gms/day) was 20.6%. No significant difference was seen in the prevalence of ever-smoker, central obesity, T2DM, hypertension, dyslipidemia, and colorectal cancer in first-degree relatives between the 2 groups. There was no significant difference between the 2 groups in FIB-4 index, hemoglobin A1c (HbA1c) level, and serum levels of HDL cholesterol, triglycerides, and CRP (Table 1).

### 2.2. Logistic Regression Analysis for Colorectal Adenoma

We analyzed 6 factors including MAFLD, NAFLD, age, sex, alcohol intake, and smoking in the stepwise procedure. Although MAFLD and age were selected as explanatory variables for the logistic regression analysis, MAFLD was identified as the only independent factor associated with colorectal adenoma (odds ratios (OR) 3.191; 95% confidence intervals (CI) 1.494–7.070; *p* = 0.003; Figure 2A). On the other hand, NAFLD was not an independent factor associated with colorectal adenoma.

### 2.3. Decision-Tree Analysis for Colorectal Adenoma

In a decision-tree algorithm, MAFLD was identified as the most important classifier for the presence of colorectal adenoma. Colorectal adenoma was observed in 51% of subjects with MAFLD, while colorectal adenoma was observed in 25% of subjects in the absence of MAFLD (Figure 2B). The second and third classifiers were age and sex, respectively. Colorectal adenoma was observed in 86% of subjects with MAFLD, ≥63 years old, and men. On the other hand, NAFLD was not identified as a classifier for the presence of colorectal adenoma.

### 2.4. Random Forest Analysis for Colorectal Adenoma

In a random forest analysis for the presence of colorectal adenoma, MAFLD was identified as the distinguishing factor with the highest variable importance. NAFLD, age, and sex were the second, third, and fourth distinguishing factors, respectively (Figure 2C).

### 2.5. Patients’ Characteristics among Subtypes of MALFD

Patients’ characteristics for subtypes of MALFD are summarized in Table 2. There was no significant difference in age, sex, and BMI among the obese-MAFLD, non-obese MAFLD, and T2DM-MAFLD groups. No significant difference was seen in the prevalence of daily alcohol intake, ever-smoker, central obesity, hypertension, dyslipidemia, and colorectal cancer in first-degree relatives among the 3 groups. In the T2DM-MAFLD group, HbA1c level was significantly higher than that in the obese and non-obese MAFLD groups. There was no significant difference in FIB-4 index, serum HDL cholesterol, triglycerides, and CRP levels among the 3 groups (Table 2).

### 2.6. Logistic Regression Analysis for Colorectal Adenoma Using Subtypes of MALFD

We analyzed 7 factors, namely, obese-MAFLD, non-obese MAFLD, T2DM-MAFLD, age, sex, alcoholic intake, and smoking in the stepwise procedure. In the stepwise procedure, non-obese MAFLD was only selected as an explanatory variable for the logistic regression analysis. Non-obese MAFLD was identified as the only independent factor associated with colorectal adenoma (OR 3.351; 95% CI 1.589–7.262; *p* ≤ 0.001; Figure 3A). On the other hand, obese- and T2DM-MALFD were not independent factors associated with colorectal adenoma.

### 2.7. Decision-Tree Analysis for Colorectal Adenoma Using Subtypes of MALFD

In a decision-tree algorithm, non-obese MAFLD was identified as the most important classifier for the presence of colorectal adenoma. Colorectal adenoma was observed in 54% of subjects with MAFLD, while colorectal adenoma was observed in 26% of subjects with obese- or T2DM-MAFLD (Figure 3B). The second and third classifiers were sex and age, respectively. Colorectal adenoma was observed in 86% of subjects with non-obese MAFLD, men, and ≥57 years old. On the other hand, both obesity and T2DM-MAFLD were not identified as a classifier for the presence of colorectal adenoma.

### 2.8. Random Forest Analysis for Colorectal Adenoma

In a random forest analysis for the presence of colorectal adenoma in MAFLD, only non-obese MAFLD was identified as a distinguishing factor with the highest variable importance. Age and obese MAFLD were the second and third factors, respectively (Figure 3C).

## 3. Discussion

In this study, we demonstrate that MAFLD is the only independent factor associated with the presence of colorectal adenoma. We also show that MAFLD, but not NAFLD, is the most important factor associated with the presence of colorectal adenoma in decision-tree and random forest analyses. Furthermore, non-obese MAFLD was associated with the presence of colorectal adenoma among the three subtypes of MAFLD.

In this study, the prevalence of NAFLD was 46.8% among subjects who underwent colonoscopy examination (58/124). The prevalence of NAFLD has been reported to be between 40 and 52% in subjects who undergo colonoscopy [13,14,15]. Metabolic disorders including NAFLD are risk factors for colorectal adenoma [16,17]. In this study, the prevalence of colorectal adenoma was 50.0%. Wong et al. reported the prevalence of colorectal adenoma as 51.6% among subjects who underwent colonoscopy examination [9]. Furthermore, systematic reviews and meta-analyses reported that the prevalence of colorectal adenoma ranged from 30 to 50% in asymptomatic adults [8,18]. Thus, in this study, the characteristics of enrolled patients with NAFLD and the prevalence of colorectal adenoma were similar to previous reports.

MAFLD was an independent factor for colorectal adenoma in this study. Decision-tree and random forest analyses also revealed that MAFLD, but not NAFLD, was the most important factor for the presence of colorectal adenoma. Recently, several studies have shown the superiority of MAFLD over NAFLD for the identification of patients with significant hepatic fibrosis, cardiovascular event, and chronic kidney disease [11,12,19,20]. Moreover, we now demonstrate the superiority of MAFLD over NAFLD to identify patients with colorectal adenoma. Although the reason for the superiority of MAFLD remains unclear, a possible explanation is a difference in the diagnostic criteria between the two definitions. NAFLD can be diagnosed regardless of the presence of metabolic dysregulation [21]. However, the presence of metabolic dysfunction is a necessary inclusion criterion for the diagnosis of MAFLD [10,22]. Metabolic dysfunction including obesity and T2DM are well-known risk factors for colorectal adenoma [23,24]. Furthermore, the diagnosis of MAFLD is independent of alcoholic intake, which is also a risk factor for colorectal adenoma [25]. Thus, MAFLD includes risk factors for colorectal adenoma, and, therefore, MAFLD may identify patients with colorectal adenoma better than NAFLD.

MAFLD consists of three subtypes, namely, obese-MAFLD, non-obese MAFLD, and T2DM-MAFLD [10,22]. We found that non-obese MAFLD was the independent factor for the presence of colorectal adenoma, while obese-MAFLD and T2DM-MAFLD were not. There was no significant difference in the number of complicating metabolic abnormalities or in the alcohol intake among the three subgroups of MAFLD. Thus, it remains unclear which factor is responsible for colorectal adenoma in the non-obese MAFLD group. Recently, non-obese NAFLD has been reported as a major subtype of NAFLD [26]. Sarcopenia and alterations in gut microbiota are pathophysiological factors associated with the development of non-obese NAFLD [27,28]. These factors are also known as risks for colorectal adenoma [29,30,31]. In addition, the transmembrane 6 superfamily member 2 (TM6SF2) gene polymorphism is associated with lean NAFLD [32], while it is also reported to be associated with colorectal adenoma [25]. Taken together, possible factors for an association between non-obese MAFLD and colorectal adenoma include alterations in skeletal muscle mass, gut microbiota, and/or TM6SF2 gene polymorphism.

There are several limitations to this study. First, this study is retrospective, with a small sample size. Second, all participants were of Asian ancestry. Third, we did not evaluate factors associated with colorectal cancer, including dietary habits and physical activity. Fourth, since all subjects were health check-up examinees, the prevalence of severe steatosis and advanced fibrosis was low in this cohort. Therefore, we could not evaluate the impact of severity of hepatic steatosis and fibrosis on the prevalence of colorectal adenomas. Fifth, we also could not evaluate the impact of MAFLD on colorectal cancer, because of the small number of patients with colorectal cancer in this cohort (2/126). Accordingly, further studies should be designed in an international prospective study with a large sample size to more comprehensively evaluate the effects of MAFLD on the development of colorectal adenomas/adenocarcinomas using various factors including lifestyle habits and hepatic steatosis/fibrosis using FibroMax indexes [33,34,35].

In conclusion, we found that MAFLD was the only independent factor associated with the presence of colorectal adenoma. In both decision-tree and random forest analyses, these results were confirmed, while NAFLD was not an independent factor. In addition, we identified that non-obese MAFLD was associated with the presence of colorectal adenoma among the three subtypes of MAFLD. Since the identification of colorectal adenoma is an effective strategy to reduce the death rate from colorectal cancer, we suggest that colonoscopy examination is better considered in patients with MAFLD, particularly those who are non-obese.

## 4. Materials and Methods

### 4.1. Study Design and Ethics

This study was designed as a multicenter cross-sectional retrospective study in Japan. The protocol conformed to the ethical guidelines of the 1975 Declaration of Helsinki as reflected by prior approval from the institutional review board of Kurume University School of Medicine (ID 20114). This research was performed in accordance with relevant guidelines and regulations. An opt-out approach was used to obtain informed consent from patients, and personal information was protected during data collection.

### 4.2. Study Population and Selection of Patients for Analysis

We enrolled health check-up examinees who met the following inclusion criteria at the Kurume University Hospital, Kumamoto Central Hospital, and Kurate Hospital in Japan from April 2018 to March 2020 (*n* = 130): (1) subjects who were 20 years old or greater, and (2) subjects who underwent colonoscopy examination from 2018 to 2020. Of these, we have excluded subjects with (1) incomplete colonoscopy examination (*n* = 1), (2) positive results of hepatitis B virus surface antigen (*n* = 1) or hepatitis C virus antibody (*n* = 2), and (3) colorectal cancer (*n* = 2). Thus, a total of 124 subjects were analyzed in this study (Figure 4). All patients were Asian. Colonoscopy examination was performed as a part of their clinical review. We excluded patients with colorectal cancer (*n* = 2) or with an incomplete examination, defined as an endoscope not reaching the cecum as documented by a picture of the ileocecal valve (*n* = 1). Finally, 124 health check-up examinees were analyzed in this study.

### 4.3. Data Collection

All data were collected retrospectively from medical records at the time of colonoscopy. The following information was obtained using a self-reported questionnaire: age, sex, comorbidity, medication use, and colorectal cancer in first-degree relatives. At the clinical review, we obtained the following data: BMI, waist circumference, blood pressure, presence or absence of T2DM, hypertension, and dyslipidemia; these were diagnosed according to standard criteria [10,36,37,38]. We obtained the data for current alcohol intake and smoking. Daily alcohol intake habit was defined as men ≥30 gms/day or women ≥20 gms/day.

### 4.4. Biochemical Analysis

Patients fasted overnight before collection of blood samples for the following tests: complete blood count, aspartate aminotransferase, alanine aminotransferase, lactate dehydrogenase, alkaline phosphatase, γ-glutamyl transpeptidase, albumin, total bilirubin, HDL cholesterol, triglycerides, fasting glucose, HbA1c, and C-reactive protein. FIB-4 index was calculated using age, serum levels of AST, ALT, and platelet count as previously described [39].

### 4.5. Diagnosis of Fatty Liver and Assessment for the Severity of Steatosis and Hepatic Fibrosis

The diagnosis of fatty liver was based on the presence of any of the following findings on abdominal ultrasonography: increased hepato-renal contrast, increased echogenicity of liver parenchyma, unclear visualization of the intrahepatic vessels, and/or impaired visualization of the diaphragm as previously described [40].

The severity of hepatic steatosis was assessed by fatty liver index as previously described [41]. The severity of hepatic fibrosis was assessed by FIB-4 index and NAFLD fibrosis score as previously described [12].

### 4.6. Diagnosis of NAFLD

The diagnosis of NAFLD was according to the EASL-EASD-EASO and American Association for the Study of Liver Diseases Clinical Practice Guidelines for the Management of NAFLD [42,43]: (1) fatty liver by abdominal ultrasonography, (2) alcohol intake no greater than 30 gms/day for men and 20 gms/day for women, and (3) no competing etiologies for fatty liver or coexisting causes of chronic liver disease [42,43].

### 4.7. Diagnosis of MAFLD

MAFLD was diagnosed according to the criteria proposed in 2020 by an international expert panel [10]. The criteria include evidence of fatty liver, in addition to one of the following: obesity, presence of T2DM, or non-obesity with evidence of metabolic dysregulation. Since all patients were Asian, BMI and waist circumstance were evaluated using cut-off values for Asians [10]. Obesity was defined as BMI ≥23 kg/m^2^ in this Asian cohort and T2DM was defined as HbA1c ≥6.5% or specific drug treatment. Metabolic dysregulation was defined as the presence of at least two metabolic risk abnormalities: (1) Central obesity (waist circumstance ≥90/80 cm in men and women), (2) blood pressure ≥130 mmHg or specific drug treatment, (3) plasma triglycerides ≥150 mg/dL or specific drug treatment, (4) plasma HDL-cholesterol <40 mg/dL for men and <50 mg/dL for women or specific drug treatment, and (5) prediabetes (fasting glucose levels 100 to 125 mg/dL or HbA1c 5.7–6.4%) [10]. Although the homeostasis model assessment of insulin resistance score and plasma high-sensitivity C-reactive protein level are metabolic risk abnormalities in the MAFLD criteria [10], these were not available in our dataset.

### 4.8. Colonoscopy Examination and Diagnosis of Colorectal Adenoma

Colonoscopy examination (CF H260AZI/PCF H290ZI; Olympus, Tokyo, Japan) was performed by endoscopists with experience of performing more than 1000 procedures. Subjects were given polyethylene glycol for bowel preparation according to the manufacturer’s instructions. A complete examination was defined as an endoscope reaching the cecum as documented by a picture of the ileocecal valve. All identified colorectal polyps were removed and were tubular adenomas, which were diagnosed by pathological findings. This is a retrospective study and diagnosis of colorectal adenoma had already been made at the entry of this study.

In all colonoscopy examinations, the quality of bowel preparation was graded as good (no or small volume of clear liquid, with >95% of the surface seen). The withdrawal time of the colonoscopy procedure was at least 6 min to minimize the chance of missing lesions [44]. Incomplete examinations were excluded from the analysis.

### 4.9. Statistical Analysis

Continuous variables are expressed as median and range or number. Categorical variables are expressed as frequencies and percentages. The differences between groups were analyzed using the Wilcoxon rank-sum test for continuous variables and the Fisher’s exact test for categorical variables. A logistic regression model was used to identify independent factors associated with the presence of colorectal adenoma. Data are expressed as OR and 95% CI.

A decision-tree algorithm was constructed to reveal profiles associated with the presence of colorectal adenoma as previously described [12,40]. A random forest analysis was used to identify factors that distinguished for the presence of colorectal adenoma as previously described [45,46]. The variable importance value which reflects the relative contribution of each variable to the model was estimated by randomly permuting its values and recalculating the predictive accuracy of the model. *p* < 0.05 was considered to indicate statistical significance. Data were analyzed using the JMP Pro15 (SAS Institute Inc., Cary, NC, USA).

## 5. Conclusions

MAFLD, particularly non-obese MAFLD, is the most important factor associated with the presence of colorectal adenoma rather than NAFLD. Colonoscopy examination should be considered in patients with MAFLD, especially those who are non-obese.

## Figures and Tables

**Figure 1 ijms-22-05462-f001:**
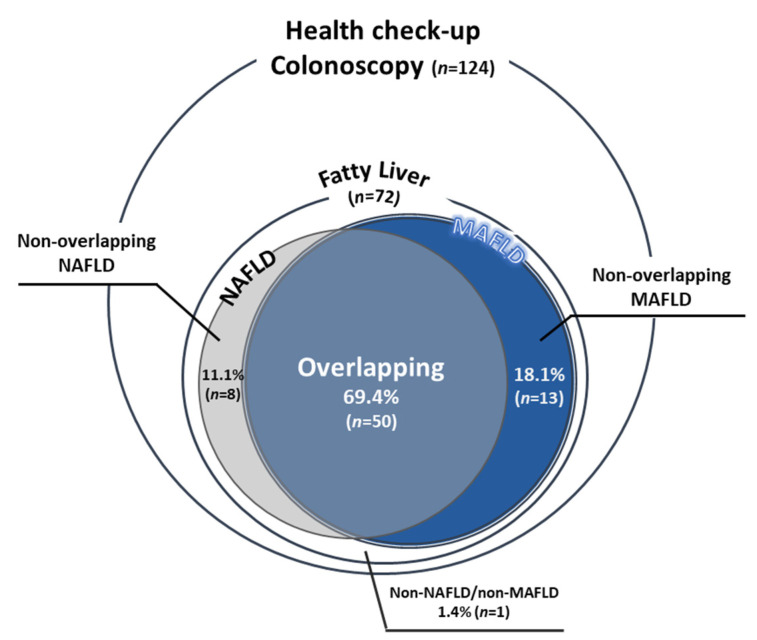
The population of MAFLD and NAFLD. The Venn diagram indicates the proportion of patients with NAFLD (gray) and patients with MAFLD (blue).

**Figure 2 ijms-22-05462-f002:**
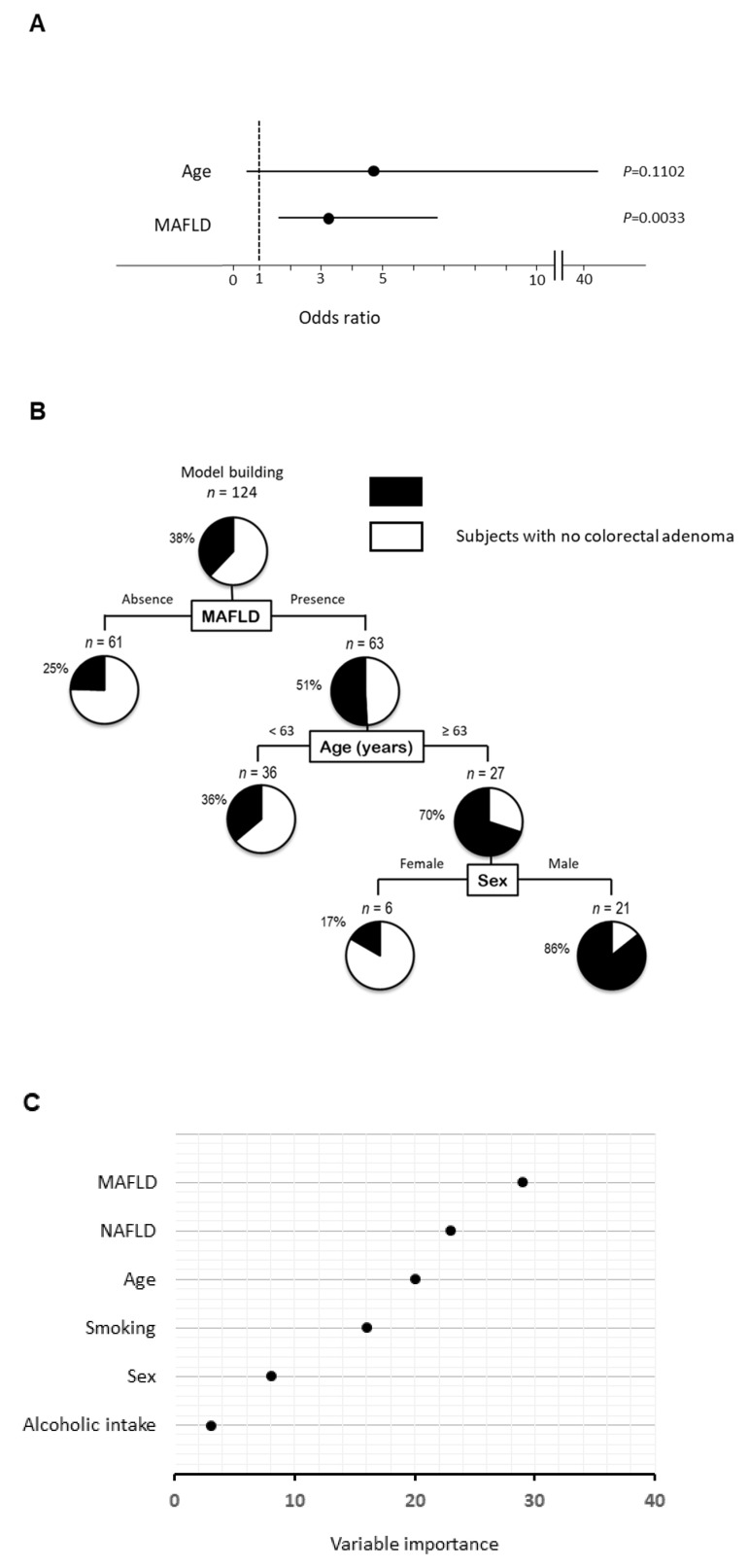
Independent factors and profiles associated with the presence of colorectal adenoma. (**A**) Independent factors for the presence of colorectal adenoma analyzed by logistic regression analysis, (**B**) profiles for the presence of colorectal adenoma analyzed by decision-tree analysis. The pie graphs indicate the proportion of patients with colorectal adenoma (black) and patients with no colorectal adenoma (white), (**C**) distinguishing factors for the presence of colorectal adenoma analyzed by random forest analysis. The relative contributions of each variable to the presence of colorectal adenoma is expressed by variable importance.

**Figure 3 ijms-22-05462-f003:**
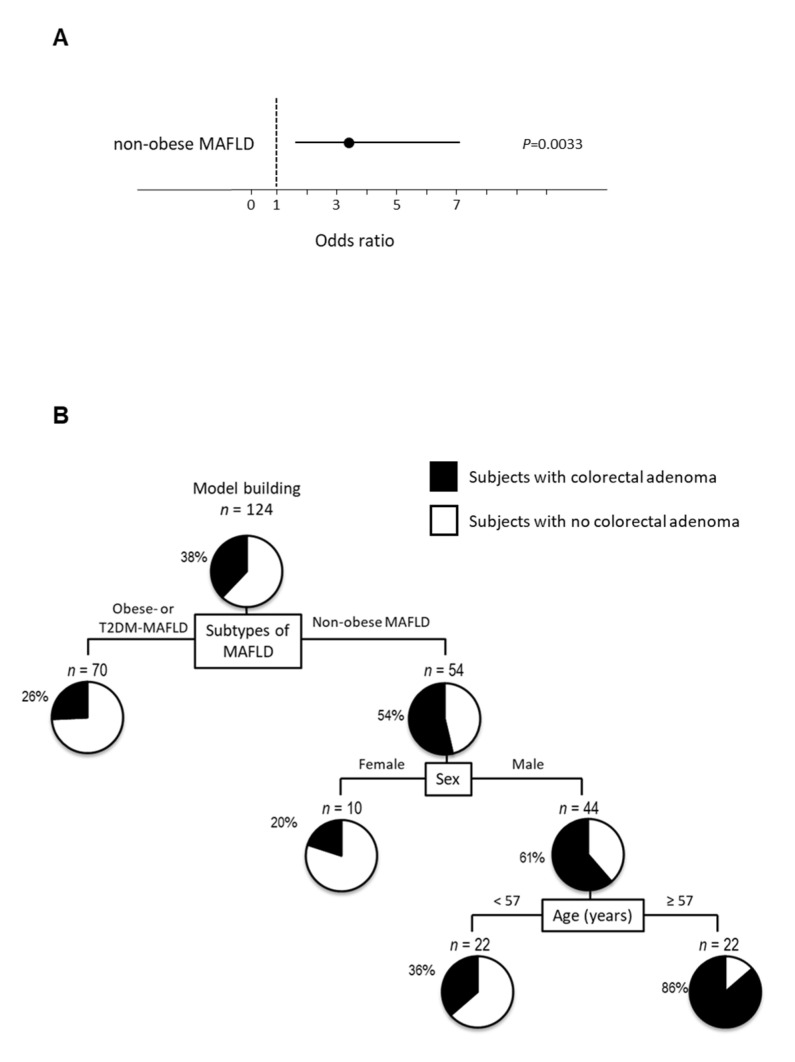
Independent factors and profiles associated with the presence of colorectal adenoma according to subtypes of MAFLD. (**A**) Independent factors for the presence of colorectal adenoma analyzed by logistic regression analysis, (**B**) profiles for the presence of colorectal adenoma analyzed by decision-tree analysis. The pie graphs indicate the proportion of patients with colorectal adenoma (black) and patients with no colorectal adenoma (white), (**C**) distinguishing factors for the presence of colorectal adenoma analyzed by random forest analysis. The relative contributions of each variable to the presence of colorectal adenoma is expressed by variable importance.

**Figure 4 ijms-22-05462-f004:**
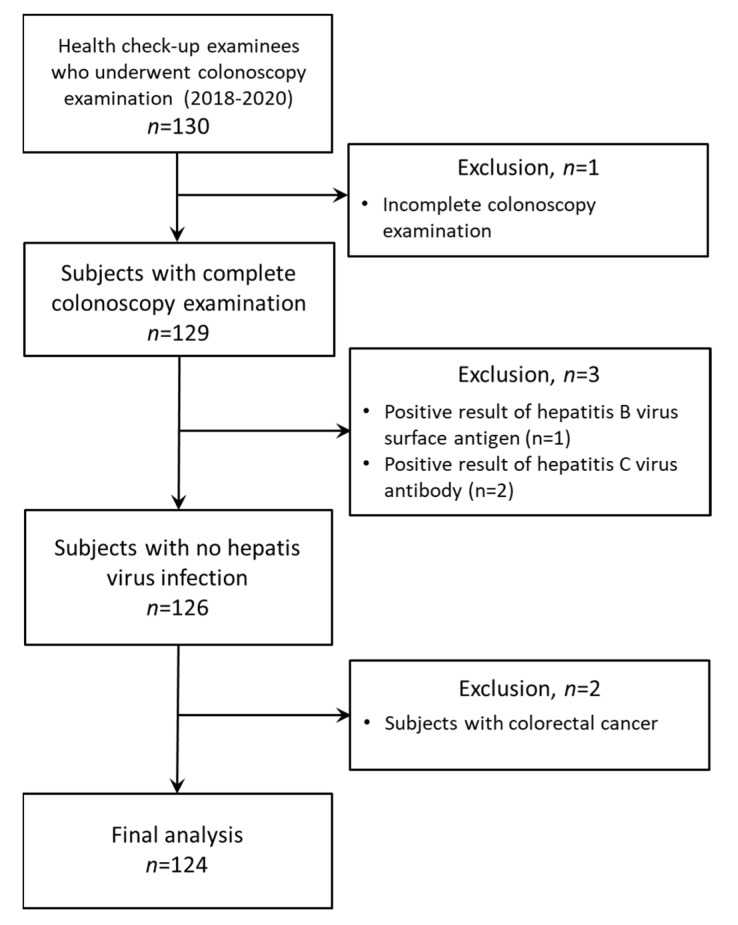
A flow chart for study populations.

**Table 1 ijms-22-05462-t001:** Patients’ characteristics.

	All Subjects		NAFLD		MAFLD		*p*
	Median (IQR)	Range (min–max)	Median (IQR)	Range (min–max)	Median (IQR)	Range (min–max)	
Number	124	N/A	46.8% (58/124)	N/A	50.8% (63/124)	N/A	N/A
Age (years)	59 (50–65)	32–88	58 (48–65)	32–80	61 (50–65)	32–80	0.5178
Sex (female/male)	19.4%/80.6% (24/100)	N/A	17.2%/82.8% (10/48)	N/A	17.5%/82.5% (11/52)	N/A	0.9747
Body mass index(kg/m^2^)	23.1 (21.0–25.7)	16.9–33.2	24.5 (21.8–26.5)	16.9–33.2	25.0 (23.0–26.8)	18.6–33.2	0.3026
Obesity (Yes/No)	52.4%/47.6% (65/59)	N/A	67.2%/32.8% (39/19)	N/A	76.2%/23.8% (48/15)	N/A	0.2739
Daily alcoholic intake (men ≥30 gms, women ≥20 gms) (Yes/No)	15.3%/84.7% (19/105)	N/A	0%/100% (0/58)	N/A	20.6%/79.4% (13/50)	N/A	0.0003
Ever-smoker	27.4% (34/124)	N/A	25.9% (15/58)	N/A	27.0% (17/63)	N/A	0.8888
Central obesity (Yes/No)	37.1%/62.9% (46/78)	N/A	50.0%/50.0% (29/29)	N/A	57.1%/42.9% (36/27)	N/A	0.4312
Systolic blood pressure (mmHg)	123 (115–132)	88–170	126 (117–136)	88–170	127 (118–137)	88–170	0.6123
Type 2 diabetes mellitus (Presence/Absence)	17.7%/82.3% (22/102)	N/A	13.8%/86.2% (8/50)	N/A	20.6%/79.4% (13/50)	N/A	0.3208
Hypertension (Presence/Absence)	25.0%/75.0% (31/93)	N/A	25.9%/74.1% (15/43)	N/A	33.3%/66.7% (21/42)	N/A	0.3691
Dyslipidemia (Presence/Absence)	17.7%/82.3% (22/102)	N/A	19.0%/81.0% (11/47)	N/A	23.8%/76.2% (15/48)	N/A	0.5169
Colorectal cancer in first-degree relatives	3.2% (4/120)	N/A	3.6% (2/58)	N/A	4.8% (3/63)	N/A	0.7168
Fatty liver	58.1% (72/124)	N/A	100% (58/58)	N/A	100% (63/63)	N/A	N/A
Fatty liver index (≤60/>60)	81%/19% (101/23)	N/A	74%/26% (43/15)	N/A	65%/35% (41/22)	N/A	0.3895
NAFLD fibrosis score (F0-F2/indeterminant score/ F3-F4)	66%/32%/2% (82/40/2)	N/A	72%/28%/0% (42/16/0)	N/A	71%/27%/2% (45/17/1)	N/A	0.6285
FIB-4 index	1.24 (0.97–1.72)	0.20– 3.67	0.98 (0.71–1.34)	0.30–3.79	0.99 (0.74–1.37)	0.30–3.79	0.5602
Platelet count (×104/µL)	22 (19–25)	10–85	23 (20–27)	15–85	24 (20–27)	12–85	0.9200
AST (U/L)	23 (19–27)	14–171	23 (20–26)	15–77	23 (20–26)	15–77	0.8467
ALT (U/L)	22 (16–30)	9–366	23 (18–31)	9–107	23 (18–32)	11–107	0.6771
Lactate dehydrogenase (U/L)	167 (151–189)	119–268	165 (147–190)	126–238	162 (147–189)	132–238	0.9966
ALP (U/L)	214 (175–260)	117–422	211 (175–270)	119–422	213 (182–265)	132–422	0.9820
GGT (U/L)	25 (19–51)	11–281	28 (21–52)	12–223	35 (22–64)	12–281	0.2765
Albumin (g/dL)	4.4 (4.2–4.6)	3.4–5.1	4.5 (4.2–4.6)	3.9–5.1	4.5 (4.2–4.7)	3.4–5.1	0.6689
Total bilirubin (mg/dL)	0.8 (0.6–1.0)	0.3–2.0	0.9 (0.6–1.0)	0.3–2	0.9 (0.7–1.0)	0.3–2.0	0.9380
HDL cholesterol (mg/dL)	59 (47–70)	28–209	56 (45–73)	33–99	56 (45–70)	33–93	0.7070
Triglycerides (mg/dL)	101 (72–150)	31–760	137 (89–167)	31–440	142 (89–199)	31–760	0.3523
Fasting glucose (mg/dL)	101 (94–107)	67–230	101 (95–111)	83–230	103 (96–129)	67–230	0.7853
HbA1c (%)	5.5 (5.4–5.9)	5.0–8.8	5.6 (5.4–5.9)	5.0–8.8	5.6 (5.4–6.0)	5.0–8.8	0.7161
CRP (mg/dL)	0.07 (0.04–0.13)	0.03– 0.31	0.09 (0.05–0.13)	0.01–0.31	0.09 (0.06–0.13)	0.01–0.31	0.7737
Colorectal adenomas (Presence/Absence)	37.9%/62.1% (47/77)	N/A	50.0%/50.0% (29/29)	N/A	50.8/49.2 (32/31)	N/A	0.9305

Data are expressed as median (interquartile range (IQR)), range, or number. Abbreviations: N/A, not applicable; NAFLD, non-alcoholic fatty liver disease; MAFLD, metabolic associated fatty liver disease; FIB-4, fibrosis-4; AST, aspartate aminotransferase; ALT, alanine aminotransferase; ALP, alkaline phosphatase; GGT, gamma-glutamyl transpeptidase; HDL cholesterol, high-density lipoprotein cholesterol; LDL cholesterol, low-density lipoprotein cholesterol; HbA1c, hemoglobin A1c; CRP, C-reactive protein.

**Table 2 ijms-22-05462-t002:** Patients’ characteristics in subtypes of MAFLD.

	Obese-MAFLD		Non-Obese MAFLD		T2DM-MAFLD		*p*
	Median (IQR)	Range (min–max)	Median (IQR)	Range (min–max)	Median (IQR)	Range (min–max)	
Number	38.7% (48/124)	N/A	43.5% (54/124)	N/A	10.5% (13/124)	N/A	N/A
Age (years)	60 (50–65)	32–80	62 (51–65)	41–80	65 (60–71)	43–73	0.2253
Sex (female/male)	14.6%/85.4% (7/41)	N/A	18.5%/81.5% (10/44)	N/A	23.1%/76.9% (3/10)	N/A	0.7394
Body mass index (kg/m^2^)	26.0 (24.8–27.0)	23.0–33.2	25.2 (22.6–27)	18.6–33.2	25.2 (23.7–26.5)	20.0–33.2	0.1800
Daily alcoholic intake (men ≥30 gms, women ≥20 gms) (Yes/No)	16.7%/83.3% (8/40)	N/A	18.5%/81.5% (10/44)	N/A	30.8%/69.2% (4/9)	N/A	0.5118
Ever-smoker	25.0% (12/48)	N/A	27.8% (15/54)	N/A	23.1% (3/13)	N/A	0.9183
Central obesity (Yes/No)	66.7%/33.3% (32/16)	N/A	64.8%/35.2% (35/19)	N/A	53.9%/46.2% (7/6)	N/A	0.6898
Systolic blood pressure (mmHg)	127 (117–141)	88–170	130 (120–140)	103–170	129 (117–141)	110–170	0.7313
Type 2 diabetes mellitus (Presence/Absence)	22.9%/77.1% (11/37)	N/A	16.7%/83.3% (9/45)	N/A	100.0%/0.0% (13/0)	N/A	<.0001
Hypertension (Presence/Absence)	31.3%/68.8% (15/33)	N/A	38.9%/61.1% (21/33)	N/A	53.9%/46.2% (7/6)	N/A	0.3121
Dyslipidemia (Presence/Absence)	20.8%/79.2% (10/38)	N/A	25.9%/74.1% (14/40)	N/A	53.9%/46.2% (7/6)	N/A	0.0574
Colorectal cancer in first-degree relatives	6.3% (3/48)	N/A	3.7% (2/54)	N/A	15.4% (2/13)	N/A	0.2858
Fatty liver index (≤60/>60)	54%/46% (26/22)	N/A	63%/37% (34/20)	N/A	69%/31% (9/4)	N/A	0.4559
NAFLD fibrosis score (F0–F2/indeterminant score/ F3-F4)	71%/27%/2% (34/13/1)	N/A	74%/24%/2% (40/13/1)	N/A	15%/77%/8% (2/10/1)	N/A	0.0019
FIB-4 index	1.15 (0.91–1.56)	0.53–3.67	1.16 (0.90–1.53)	0.20–3.67	1.81 (1.36–2.01)	0.53–2.90	0.1427
Platelet count (×10^4^/µL)	22 (18–27)	12–37	25 (20–27)	12–85	18 (16–24)	12–37	0.2141
AST (U/L)	23 (20–28)	16–77	23 (20–27)	15–77	25 (21–26)	17–42	0.8104
ALT (U/L)	27 (20–38)	11–107	26 (19–34)	13–107	23 (18–30)	14–44	0.3305
Lactate dehydrogenase (U/L)	162 (146–185)	132–238	166 (149–196)	132–238	174 (157–218)	145–238	0.1820
ALP (U/L)	203 (174–265)	132–422	213 (186–264)	137–422	238 (165–337)	144–422	0.4717
GGT (U/L)	43 (24–73)	13–281	40 (24–68)	12–281	24 (17–54)	14–151	0.6126
Albumin (g/dL)	4.5 (4.2–4.7)	3.9–5.1	4.5 (4.2–4.7)	3.4–5.1	4.3 (4.0–4.7)	3.2–4.9	0.3471
Total bilirubin (mg/dL)	0.9 (0.6–1.0)	0.3–2.0	0.8 (0.7–1.0)	0.3–2.0	0.9 (0.8–1.1)	0.6–1.1	0.9535
HDL cholesterol (mg/dL)	54 (43–63)	33–92	56 (43–74)	33–93	55 (46–65)	38–82	0.4955
Triglycerides (mg/dL)	146 (105–248)	31–440	141 (89–202)	34–760	139 (84–182)	73–352	0.8513
Fasting glucose (mg/dL)	103 (97–112)	67–230	103 (96–111)	67–141	125 (108–148)	67–230	0.0003
HbA1c (%)	5.7 (5.4–6.1)	5.0–8.8	5.6 (5.4–5.9)	5.0–7.1	6.5 (6.2–7.0)	5.2–8.8	<0.0001
CRP (mg/dL)	0.10 (0.06–0.14)	0.03–0.31	0.10 (0.06–0.13)	0.01–0.31	0.10 (0.06–0.13)	0.04–0.29	0.8466
Colorectal adenomas (Presence/Absence)	54.2%/45.8% (26/22)	N/A	53.7/46.3 (29/25)	N/A	38.5%/61.5% (5/8)	N/A	0.5750

Note. Data are expressed as median (interquartile range (IQR)), range, or number. Abbreviations: N/A, not applicable; NAFLD, non-alcoholic fatty liver disease; MAFLD, metabolic associated fatty liver disease; FIB-4, fibrosis-4; AST, aspartate aminotransferase; ALT, alanine aminotransferase; ALP, alkaline phosphatase; GGT, gamma-glutamyl transpeptidase; HDL cholesterol, high-density lipoprotein cholesterol; LDL cholesterol, low-density lipoprotein cholesterol; HbA1c, hemoglobin A1c; CRP, C-reactive protein.

## Data Availability

The data presented in this study are available on request from the corresponding author.

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
