# Peer review of "Non-Obese MAFLD Is Associated with Colorectal Adenoma in Health Check Examinees: A Multicenter Retrospective Study"

_ijms, 2021, doi:10.3390/ijms22115462_

Round 1

Reviewer 1 Report

This study identified a link between development of colorectal adenomas with MAFLD in non-obese patients. The manuscript is very nicely written and structured. It definitely contains important information for further development of the field. Moreover, all study design and its description are technically sound and are based on current consensus approaches. I'd like to ask just two questions that can be important to additionally increase clearness of the study. 1. Can any control group (i.e. non-MAFLD patients undergoing colonoscopy) be used to show the rates of adenocarcinomals? 2. Since the patients underwent quite extensive characterization, can Fibromax Indexes to show the degree of NALFD/fibrosis/steatosis be calculated?

Author Response

To REVIEWER 1

Thank you very much for your letter regarding our manuscript (ijms-1211641). We appreciate your comments, which have helped us to improve our manuscript. In line with your comments, please find below our response.

Comment 1: Can any control group (i.e. non-MAFLD patients undergoing colonoscopy) be used to show the rates of adenocarcinomas?

Answer: We appreciate your pointing it out. As you suggested, we agree that it is important to show the rate of adenocarcinomas in the control group (non-MAFLD patients undergoing colonoscopy). In our cohort, only 2 subjects had colorectal adenocarcinoma and they were fulfilled with MAFLD criteria. However, the number of patients with colorectal adenocarcinomas was very small and was insufficient to investigate an association between MAFLD and the prevalence of adenocarcinoma. Therefore, we added this issue as a limitation of this study (line 219-220, line 249).

Comment 2: Since the patients underwent quite extensive characterization, can Fibromax Indexes to show the degree of NALFD/fibrosis/steatosis be calculated?

Answer: We appreciate your valuable comment. As you pointed out, it is important to evaluate the degree of NALFD/fibrosis/steatosis using FibroMax™ [1-3]. However, we do not have the data for the following parameters for the assessment of FibroMax™: alpha-2-macroglobulin, apolipoprotein A1, and haptoglobin. Therefore, instead of FibroMax™, we have evaluated fatty liver index and NAFLD fibrosis score for the assessment of the degree of steatosis and fibrosis, respectively. There was no significant difference in the fatty liver index and NAFLD fibrosis score between the NAFLD and MAFLD groups. Since all subjects were health check-up examinees, the prevalence of severe steatosis and advanced fibrosis was low in this study. Accordingly, we could not evaluate the impact of severity of steatosis and fibrosis on colorectal adenomas. This issue was described as a limitation of this study in the revised manuscript (line 216-219, 222-224, 280-282). Again, we appreciate your comments, which have helped us to improve our manuscript.

References

1      Morra, R.; Munteanu, M.; Imbert-Bismut, F.; Messous, D.; Ratziu, V., Poynard, T. (2007) FibroMAX: towards a new universal biomarker of liver disease? Expert Rev Mol Diagn 7: 481-90.

2      Munteanu, M.; Ratziu, V.; Morra, R.; Messous, D.; Imbert-Bismut, F., Poynard, T. (2008) Noninvasive biomarkers for the screening of fibrosis, steatosis and steatohepatitis in patients with metabolic risk factors: FibroTest-FibroMax experience. J Gastrointestin Liver Dis 17: 187-91.

3      Gudowska, M.; Wojtowicz, E.; Cylwik, B.; Gruszewska, E., Chrostek, L. (2015) The Distribution of Liver Steatosis, Fibrosis, Steatohepatitis and Inflammation Activity in Alcoholics According to FibroMax Test. Adv Clin Exp Med 24: 823-7.

Reviewer 2 Report

The manuscript is a study on the association of non-obese metabolic dysfunction-associated fatty liver disease (MAFLD) with colorectal adenoma. Authors aimed to investigate the impact of MAFLD on the prevalence of colorectal adenoma by comparing it to non-alcoholic fatty liver disease (NAFLD) in health check examinees. The authors enrolled 124 consecutive health check examinees who underwent colonoscopy. NAFLD and MAFLD were present in 58 and 63 examinees. Colorectal adenoma was diagnosed by biopsy. MAFLD was identified as the only independent factor associated with the presence of colorectal adenoma.

I read the article with interest; I commend the authors for several strengths of their work, including addressing an interesting and timely topic. The manuscript is of clinical relevance.

Considering these strengths, though, I found some areas where I would have appreciated greater clarity as I read the manuscript.

  • The design of the study is incomprehensible. How is the study structured? A flow chart showing the procedure for the entire should be included.
  • What was the intended study sample? How were the subjects selected?
  • What were the inclusion and exclusion criteria?
  • At what point of the study the diagnosis of colorectal adenoma was made?
  • In how many of the studied subjects such a diagnosis was made?
  • What percentage of all studied subjects does this number constitute?
  • How does this relate to the incidence of colorectal adenoma in the entire population?
  • Which data were obtained retrospectively?

Author Response

To REVIEWER 2

Thank you very much for your letter regarding our manuscript (ijms-1211641). We appreciate your comments, which have helped us to improve our manuscript. In line with your comments, please find below our response.

Comment 1: The design of the study is incomprehensible.

Answer: We apologize that we did not provide sufficient information for the study design. We enrolled health check-up examinees who met the following inclusion criteria (n=130): Inclusion criteria 1) subjects who were 20 years old or greater, and 2) subjects who underwent colonoscopy examination from 2018 to 2020. Of these, we have excluded subjects with 1) incomplete colonoscopy examination (n=1), 2) positive results of hepatitis B virus surface antigen (n=1) or hepatitis C virus antibody (n=2), and 3) colorectal cancer (n=1). Thus, a total of 124 subjects were analyzed in this study (Figure 4). In the revised manuscript, we provided a flow chart showing the entire procedure. We also describe the patients' selection, the inclusion, and exclusion criteria in the revised manuscript (line 243-250).

Comment 2: At what point of the study, the diagnosis of colorectal adenoma was made?

Answer: We apologize for our unclear description of the diagnosis of colorectal adenomas. This is a retrospective study and diagnosis of colorectal adenoma had already been made at the entry of this study. Biopsy was performed for all polyps, and the diagnosis of colorectal adenomas was based on histology in all cases. We have added this information in the revised manuscript (line 259, 311-312).

Comment 3: Characteristics of all enrolled subjects

Answer: As you pointed out, we did not provide the data for the characteristics of all subjects. According to your suggestion, we have added the data in Table 1. In this study, 37.9% (47/124) of patients were diagnosed with colorectal adenoma. This prevalence is in good agreement with the previous studies, which reported the incidence of colorectal adenoma in the entire population [1-3]. Again, we appreciate your comments, which have helped us to improve our manuscript.

Comment 4: Which data were obtained retrospectively?

Answer: We apologize that we did not provide sufficient information for the study design. This is a retrospective study and all data were obtained from medical records. We added the description for the study design in the revised manuscript (line 235, 259, 311-312).

References

1      Dawwas, M.F. Adenoma detection rate and risk of colorectal cancer and death. N Engl J Med. 2014; 370: 2539-40, doi:10.1056/NEJMc1405329

2      Click, B.; Pinsky, P.F.; Hickey, T.; Doroudi, M., Schoen, R.E. Association of Colonoscopy Adenoma Findings With Long-term Colorectal Cancer Incidence. JAMA. 2018; 319: 2021-31, doi:10.1001/jama.2018.5809

3      Blackett, J.W.; Verna, E.C., Lebwohl, B. Increased Prevalence of Colorectal Adenomas in Patients with Nonalcoholic Fatty Liver Disease: A Cross-Sectional Study. Dig Dis. 2020; 38: 222-30, doi:10.1159/000502684

Round 2

Reviewer 2 Report

The authors have addressed satisfactorily most of the points raised by the reviewer.